# Ventral striatum dopamine release encodes unique properties of visual stimuli in mice

L Sofia Gonzalez[1,2,3†], Austen A Fisher[1,2†], Shane P D'Souza[4,5,6], Evelin M Cotella[1,2], Richard A Lang[4,5,6], J Elliott Robinson[1,2]*

[1]Rasopathy Program, Division of Experimental Hematology and Cancer Biology, Department of Pediatrics, Cincinnati Children's Hospital Medical Center, Cincinnati, United States; [2]Department of Pediatrics, University of Cincinnati College of Medicine, Cincinnati, United States; [3]Neuroscience Graduate Program, University of Cincinnati College of Medicine, Cincinnati, United States; [4]The Visual Systems Group, Abrahamson Pediatric Eye Institute, Cincinnati Children's Hospital Medical Center, Cincinnati, United States; [5]Science of Light Center, Cincinnati Children's Hospital Medical Center, Cincinnati, United States; [6]Department of Ophthalmology, University of Cincinnati College of Medicine, Cincinnati, United States

*For correspondence: elliott.robinson@cchmc.org

†These authors contributed equally to this work

**Abstract** The mesolimbic dopamine system is an evolutionarily conserved set of brain circuits that play a role in attention, appetitive behavior, and reward processing. In this circuitry, ascending dopaminergic projections from the ventral midbrain innervate targets throughout the limbic forebrain, such as the ventral striatum/nucleus accumbens (NAc). Dopaminergic signaling in the NAc has been widely studied for its role in behavioral reinforcement, reward prediction error encoding, and motivational salience. Less well characterized is the role of dopaminergic neurotransmission in the response to surprising or alerting sensory events. To address this, we used the genetically encoded dopamine sensor dLight1 and fiber photometry to explore the ability of striatal dopamine release to encode the properties of salient sensory stimuli in mice, such as threatening looming discs. Here, we report that lateral NAc (LNAc) dopamine release encodes the rate and magnitude of environmental luminance changes rather than the visual stimulus threat level. This encoding is highly sensitive, as LNAc dopamine could be evoked by light intensities that were imperceptible to human experimenters. We also found that light-evoked dopamine responses are wavelength-dependent at low irradiances, independent of the circadian cycle, robust to previous exposure history, and involve multiple phototransduction pathways. Thus, we have further elaborated the mesolimbic dopamine system's ability to encode visual information in mice, which is likely relevant to a wide body of scientists employing light sources or optical methods in behavioral research involving rodents.

## Editor's evaluation

In this manuscript, Gonzalez et al. investigated the dynamics of dopamine signals in the lateral shell of the nucleus accumbens (LNAc) in response to different types of carefully defined visual stimuli. Contrary to reigning theories of dopamine signaling, the authors presented convincing evidence that LNAcc dopamine transients tracked visual sensory transitions rather than any immediately apparent motivational variable. These important findings based on compelling evidence point to a potentially new role for dopamine signaling in the ventral striatum.

## Introduction

The mesolimbic dopamine system is an evolutionarily conserved set of circuits that plays a role in approach and avoidance, appetitive behavior, and reward processing (*Wise, 2004*; *Everitt and Robbins, 2005*; *Alcantara et al., 2022*). In this circuitry, ascending dopaminergic projections from the ventral midbrain, including the ventral tegmental area (VTA), innervate targets throughout the limbic forebrain, such as the ventral striatum/nucleus accumbens (NAc). Dopaminergic signaling in the NAc has been widely studied for its involvement in motivational salience, behavioral reinforcement, and reward prediction error encoding (*Schultz et al., 2015*; *Berridge and Robinson, 2016*; *Watabe-Uchida et al., 2017*; *Berke, 2018*). Less well characterized is the role of dopaminergic neurotransmission in the response to unpredicted or alerting sensory events, which may encourage investigation or prime motivated behavioral responses to these stimuli (*Horvitz, 2000*; *Bromberg-Martin et al., 2010a*; *Schultz, 2010*). While many previous studies have reported phasic firing of dopaminergic neurons in response to light flashes in laboratory animals (*Horvitz et al., 1997*; *Comoli et al., 2003*; *Dommett et al., 2005*), it is unclear how NAc dopamine release encodes the properties and/or emotional valence of arousing visual stimuli, such as visual threats.

Across a range of species (*Ball and Tronick, 1971*; *Sun and Frost, 1998*; *Maier et al., 2004*; *Nakagawa and Hongjian, 2010*; *Yilmaz and Meister, 2013*; *Temizer et al., 2015*), rapidly approaching objects or looming visual threats elicit automatic defensive or avoidance responses. In mice, the presentation of an expanding, overhead, black disc that simulates an aerial predator approach (a looming stimulus) promotes rapid escape to an available shelter, followed by long periods of freezing (*Yilmaz and Meister, 2013*). In our previous work published in eLife (*Robinson et al., 2019*), mice modeling cognitive dysfunction associated with neurofibromatosis type 1 (NF1) exhibited more vigorous escape in responses to looming stimulus presentation. Additionally, NAc dopamine release evoked by a white light stimulus was higher in NF1 model mice compared to wildtype littermates, which correlated with behavioral conditioning abnormalities in *Nf1* mutants. Despite the demonstration that white light can induce NAc dopamine release (*Robinson et al., 2019*; *Kutlu et al., 2021*), the striatal dopamine response to visual threats is not well characterized in mice. Additionally, it is unknown what visual stimulus characteristics – if any – are encoded by NAc dopamine. Thus, one cannot fully interpret the significance of aberrant responses in neurodevelopmental disease models without a more thorough understanding of visual stimulus encoding by mesolimbic dopamine release in typically developing subjects.

In this Research Article, we sought to probe ventral striatal dopaminergic responses to arousing visual stimuli, including looming visual threats. Given the ability of dopaminergic neurons to signal stimulus saliency (*Bromberg-Martin et al., 2010b*), we hypothesized that looming discs would induce 'alerting' NAc dopamine release whose magnitude would scale proportionately with perceived threat intensity. To test this hypothesis, we utilized the genetically-encoded sensor dLight1 (*Patriarchi et al., 2019*) to monitor dopamine release in the lateral NAc (LNAc) of freely moving adult C57Bl/6J mice with fiber photometry, as performed previously (*Robinson et al., 2019*). The LNAc was chosen given our previous observation that light stimuli evoked robust dopamine transients in this locus (*Robinson et al., 2019*). Here, we report that lateral NAc dopamine release reliably reads out unique visual stimulus properties in mice, a phenomenon that is likely relevant to a wide body of scientists employing light sources or optical methods in behavioral research.

## Results

### Dopaminergic responses to looming visual threats

To explore the encoding of visual threats by ventral striatum dopamine, we stereotaxically injected an adeno-associated viral vector (AAV5-hSyn-dLight1.2) to express dLight1.2 in the lateral nucleus accumbens of adult C57Bl/6J mice, followed by implantation of a 400 µm optical fiber for sensor excitation and emitted photon collection with fiber photometry (*Figure 1—figure supplement 1*). Our fiber photometry system utilized a 465 nm LED for sensor excitation and a 405 nm LED for isosbestic (control) excitation, which was used as a reference signal to account for the effects of photobleaching and movement artifacts (*Figure 1—figure supplement 1*). *Post hoc* histological analysis of dLight1 recording sites showed good targeting of the LNAc (*Figure 1—figure supplement 1*) with the distribution of optical fiber tip locations centered upon the medial LNAc shell (LNAcS) and spanning from

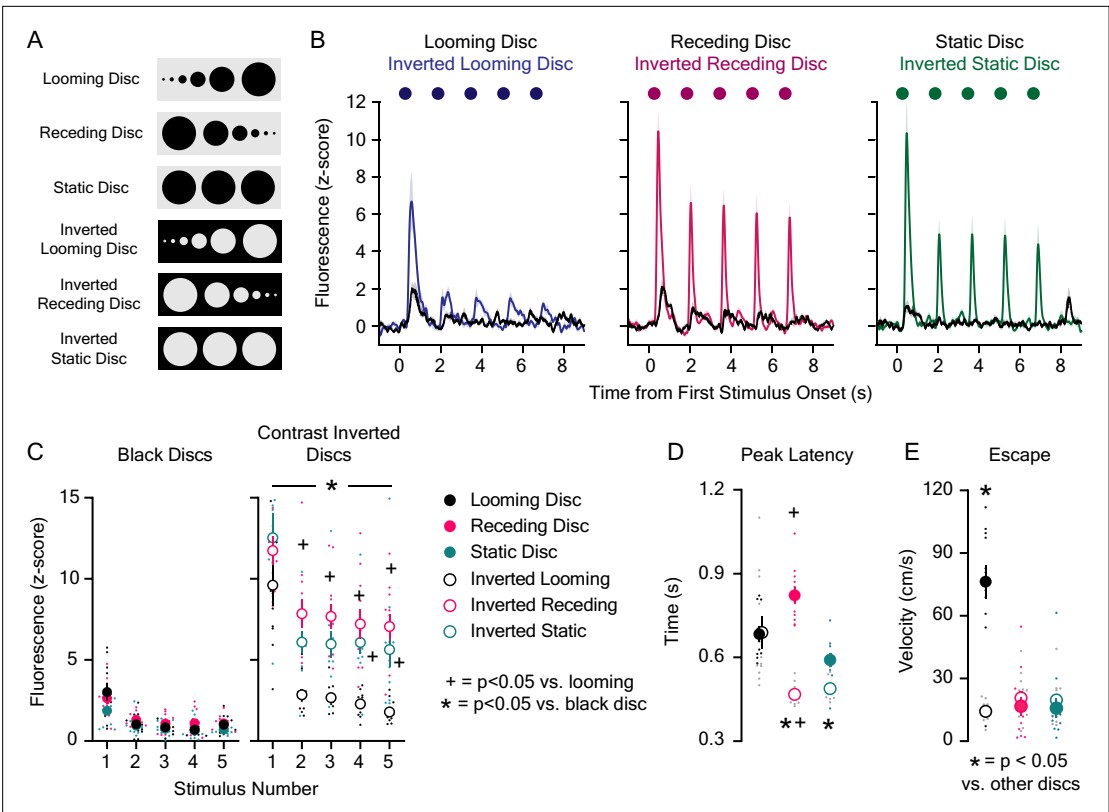

**Figure 1.** Lateral NAc (LNAc) dopaminergic encoding of visual threats. (**A**) Fluorescent dopamine signals were recorded during the presentation of black or contrast-inverted looming and control discs. (**B**) Average dLight1 response to trains of five black or contrast inverted discs ± standard error of the mean (SEM). (**C**) The dLight1 response to black or inverted discs was dependent on disc color/background, disc type (static vs. looming vs. receding), and stimulus number (n=11; 3-way repeated measures ANOVA; $F_{8,240} = 2.02$, $p_{disc\ background\ x\ disc\ type\ x\ stimulus\ number} = 0.045$; $F_{1,240} = 143.92$, $p_{background} <0.001$; $F_{2,240} = 150.32$, $p_{disc\ type} <0.001$; $F_{4,240} = 192.64$, $p_{stimulus\ number} <0.001$). Bonferroni *post hoc* tests revealed that contrast inverted discs evoked more dopamine than black discs. Contrast inverted looming discs evoked less dopamine than inverted static and receding discs after the first presentation. (**D**) dLight1 transient peak latency was dependent on disc color/background and disc type (n=11; 2-way repeated measures ANOVA; $F_{2,20} = 64.78$, $p_{disc\ background\ x\ disc\ type} <0.001$; $F_{1,20} = 25.69$, $p_{background} <0.001$; $F_{2,20} = 7.58$, $p_{disc\ type} = 0.01$). Bonferroni *post hoc* tests showed that contrast inverted static and receding discs evoked transients with shorter latency compared to black discs. Additionally, transients evoked by contrast inverted receding discs had shorter latency than contrast inverted looming discs. (**E**) Escape velocity following overhead disc presentation was dependent on disc color/background and disc type (n=12; 2-way repeated measures ANOVA; $F_{2,22} = 49.28$, $p_{disc\ background\ x\ disc\ type} <0.001$; $F_{1,22} = 18.38$, $p_{background} = 0.001$; $F_{2,22} = 28.89$, $p_{disc\ type} <0.001$). Bonferroni *post hoc* tests showed that black looming discs induced greater escape velocity than all other overhead discs. For panels C and D, * indicates p<0.05 vs. black disc of the same type (e.g. black static disc vs. contrast inverted static disc); + indicates p<0.05 vs. looming disc of the same color (e.g. black looming vs. black receding disc). For panel E, * indicates p<0.05 vs. other overhead discs.

The online version of this article includes the following source data and figure supplement(s) for figure 1:

**Source data 1.** Source data and associated statistical testing results for *Figure 1*.

**Figure supplement 1.** Overview of experimental setup and workflow.

the lateral edge of the LNAcS to the lateral aspect of the NAc core medially. This region is innervated by the broad axonal arbors of lateral VTA dopamine neurons (*Beier et al., 2015*), whose terminals respond similarly to aversive stimuli in the LNAcS and lateral NAc core in mice (*de Jong et al., 2019*). Additional details regarding the experimental setup and workflow are presented in the *Materials and methods* and *Figure 1—figure supplement 1*.

Following surgical recovery, we measured dLight1 signals evoked by looming discs (*Figure 1A–D*; *Video 1*) using a custom Bonsai-controlled (*Lopes et al., 2015*) setup for programmable visual stimulus presentation on an overhead liquid crystal display (LCD) within a light and sound-attenuating chamber. During photometry recordings, mice were exposed to trains of five overhead, black, looming discs on a light gray background that we empirically determined produce short-latency escape in C57Bl/6J mice (*Figure 1E*, *Video 2*), consistent with previous studies (*Evans et al., 2018*; *Yilmaz and Meister, 2013*). As controls, we presented mice with trains of discs that do not reliably evoke

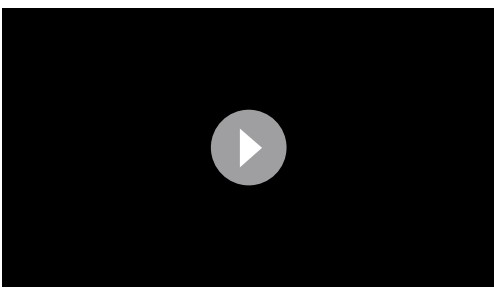

**Video 1.** Animation showing the black and contrast inverted expanding (looming), receding, and static disc stimuli.
https://elifesciences.org/articles/85064/figures#video1

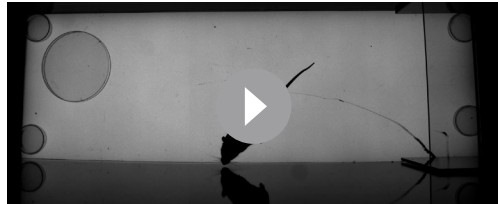

**Video 2.** The behavioral response to the presentation of black looming discs on a light background when mice entered the threat zone of a rectangular arena. https://elifesciences.org/articles/85064/figures#video2

defensive responses (*Figure 1E*), such as a static disc (a fixed 30.5 cm black disc on a light gray background), a receding disc (a black disc that contracted from 30.5 cm to 0 cm on a light gray background), and contrast inverted discs (light gray static, looming, or receding discs on a black background). We observed that looming discs induced low-amplitude dopamine transients at the onset of the first stimulus in each train that – contrary to our hypothesis – was not significantly different from the dLight1 responses to non-threatening static and receding discs (*Figure 1B–C*). Surprisingly, repeating these experiments with contrast-inverted discs that do not induce escape (*Figure 1E*) evoked ~3–6-fold greater dopamine release than black discs (*Figure 1B–C*). This raised the possibility that LNAc dopamine release tracks stimulus brightness rather than threat intensity.

## Dopaminergic responses to rapid changes in environmental lighting conditions

Because inverted looming discs, in which the number of bright overhead pixels ramps as the disc expands, produced lower amplitude (*Figure 1C*) and longer latency (*Figure 1D*) dLight1 responses than static or receding inverted discs with an instantaneous pixel change, we hypothesized that LNAc dopamine may encode the rate of change of dark-to-light transitions. To test this possibility, we exposed mice to full-screen, instantaneous transitions from black to light gray during dLight1 recordings, which eliminated disc edge motion as a contributing visual stimulus property. We found that instantaneous dark-to-light transitions produced a high amplitude ($10.38 \pm 0.43$ z-score), short duration (full width a half-maximal amplitude: $143 \pm 9.7$ ms) dopamine transient that peaked $434 \pm 3.3$ ms after transition onset (*Figure 2A*). Lengthening the dark-to-light transition time (i.e. the fade-in time) to full-screen illumination (*Figure 2B*) non-linearly decreased the magnitude of the dLight1 peak and increased the peak latency (*Figure 2C*). For transition times less than ~500 ms, the dopamine peak latency closely matched the fade-in time, above which peak response occurred hundreds of milliseconds to seconds before full field illumination was reached (*Figure 2C*). When transition times were greater than 1 s, evoked dLight1 transients were often too small to accurately resolve from the fluorescent baseline for individual mice. However, averaging the fluorescence trace from all mice prior to peak detection allowed signals to be resolved for longer transition times. Thus, results are presented as both the fluorescence peak(s) derived from the photometry trace averaged across all mice (*Figure 2C*) and individual mice (*Figure 2—figure supplement 1*), which showed high concordance for transition times of 1 s or less (*Figure 2—figure supplement 1*). No dLight1 response was reliably evoked by a 10 s dark-to-light transition despite the stimulus ramping to the same number of bright pixels as trials with shorter transition times (*Figure 2B*).

Next, we examined whether LNAc dopamine release also reads out the magnitude of environmental lighting changes by measuring the dLight1 response to 10 s, instantaneous exposures to white light across a range of intensities ($0.2 \text{ nW/cm}^2$ – $5.0 \text{ μW/cm}^2$, measured at mouse level) generated by a light emitting diode (LED) presented across 10 trials with a randomized inter-stimulus interval (ISI) between 90 and 180 s (*Figure 2D–F*). High irradiance LED illumination ($5 \text{ μW/cm}^2$) evoked a dLight1 transient at stimulus onset (*Figure 2D*) that was similar to transients evoked by the LCD monitor (*Figure 2A–B*; irradiance at mouse level: $11 \text{ μW/cm}^2$). This response did not habituate from trial-to-trial (*Figure 2—figure supplement 1*), was independent of the time of testing within the vivarium

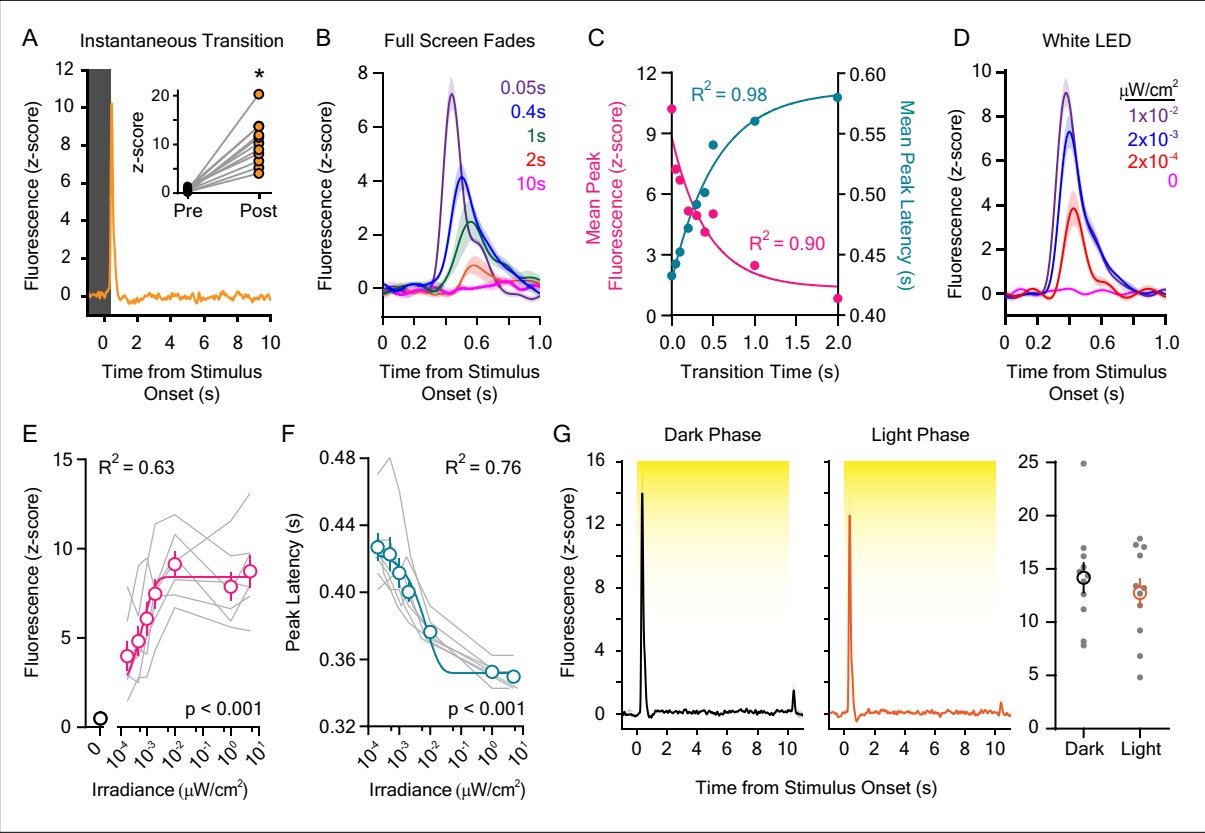

**Figure 2.** Dopaminergic responses to rapid dark-to-light transitions. (**A**) Instantaneous liquid crystal display (LCD) screen transitions from dark to light-evoked rapid dopamine release at stimulus onset when compared to the pre-stimulus baseline (inset: baseline and stimulus-induced dLight1 peak values for individual mice; n=11; paired $t$-test; $t_{10}$=7.01, p<0.001). (**B**) dLight1 responses to the onset of LCD screen dark-to-light transitions at different transition lengths (0.05–2.0 s) ± SEM. (**C**) The magnitude (*pink*) and latency (*teal*) of dopaminergic responses to dark-to-light transitions varied non-linearly depending on transition speed (peak amplitude: one-phase exponential decay, $y_0$=8.84 z-score, plateau = 1.36 z-score, tau = 0.43 s, $R^2$ = 0.90; peak latency: one-phase exponential association, $y_0$=0.43 ms, plateau = 0.59 ms, tau = 0.54 s, $R^2$ = 0.98). (**D**) dLight1 responses to the onset 10 s white light emitting diode (LED) stimuli across a range of irradiances (0 $\mu$W/cm$^2$ – 0.01 $\mu$W/cm$^2$) ± SEM. (**E**) The magnitude of the dopaminergic response to 10 s white LED stimuli was dependent on the stimulus irradiance (n=7; 1-way repeated measures ANOVA; $F_{7,42}$ = 38.79, <0.001). Data are shown with a one-phase exponential association fit ($y_0$=1.20 z-score, plateau = 0.84 z-score, tau = 0.00074 $\mu$W/cm$^2$, $R^2$ = 0.63). (**F**). The latency of the dopaminergic response to 10 s white LED stimuli was dependent on the stimulus irradiance (n=7; 1-way repeated measures ANOVA; $F_{6,36}$ = 47.35, p<0.001). Data are shown with a one-phase exponential decay fit ($y_0$=0.42 ms, plateau = 0.35 ms, tau = 0.0079 $\mu$W/cm$^2$, $R^2$ = 0.76). (**G**). The dopaminergic response to 5.0 $\mu$W/cm$^2$ white light was not different (*right*) if measured at the beginning of the vivarium dark (*left*) or light (*center*) phase of the day-night cycle (n=11; paired $t$-test; $t_{10}$=1.27, p=0.23). In all panels, * indicates p<0.05.

The online version of this article includes the following source data and figure supplement(s) for figure 2:

**Source data 1.** Source data and associated statistical testing results for *Figure 2*.

**Figure supplement 1.** Additional data: dopaminergic responses to audiovisual stimuli.

**Figure supplement 1—source data 1.** Source data and associated statistical testing results for *Figure 2—figure supplement 1*.

day-night cycle (*Figure 2G*), and was significantly larger than the response to auditory tones (80 dB; 1–16 kHz; *Figure 2—figure supplement 1*). When LED irradiance was reduced, we observed an intensity-dependent decrease in the magnitude of the dLight1 peak and an increase in the response latency (*Figure 2E–F*), consistent with Bloch's law of temporal summation in mammalian photoreceptors (*Scharnowski et al., 2007*; *Donner, 2021*). Significant dopaminergic responses were observed at all irradiances tested, including 0.2 nW/cm$^2$, which was not perceptible to the human experimenter. As a point of reference, the lock screen of a Samsung S21 smart phone on the lowest brightness setting had an irradiance of 20 nW/cm$^2$ when placed in the same position as the white LED. Likewise, time-locked dopamine release could be evoked by simply uncovering the enclosure peephole that allows users to observe mouse behavior (irradiance: 17 nW/cm$^2$; *Figure 2—figure supplement 1*). These results were not likely caused by mouse movement, as illumination of a white LED that was 1000-fold

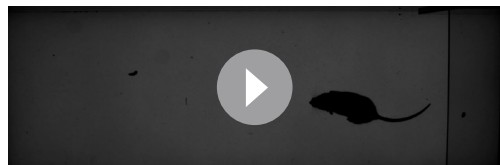

**Video 3.** The behavioral response to the illumination of a spotlight when mice entered the target zone of a rectangular arena.
https://elifesciences.org/articles/85064/figures#video3

more intense (5 mW/cm$^2$) than the highest irradiance tested had little effect on behavior when freely exploring mice entered a target zone within a dark arena (*Video 3*). Thus, LNAc dopamine release is sensitively evoked by ambient light and reliably encodes the speed of these lighting transitions over short timescales.

## Dopaminergic responses to repeated light stimuli

Previous literature suggests that dopaminergic neuron firing (*Schultz, 1998*) and dopamine release in the medial NAc core (*Kutlu et al., 2022*) in response to novel sensory events habituates as the stimulus becomes familiar. In order to test if the dLight1 response to 5 µW/cm$^2$ white LED light is affected by a repeated exposure, we exposed mice to twenty consecutive 1 s white light pulses over five trials (100 pulses total) across a range of ISIs (10 ms to 10 s; *Figure 3A*). We found that light-evoked dopamine transient magnitude decayed logarithmically as a function of the ISI duration (*Figure 3B*). When the ISI was short (e.g. 10–100 ms), dLight1 responses habituated rapidly. This is exemplified by the dopaminergic response to 40 Hz light flicker, which is used therapeutically to enhance neural activity in the context of Alzheimer's disease (*Singer et al., 2018*). Presentation of a sixty-second 40 Hz white LED flicker (5.0 µW/cm$^2$ irradiance, 50% duty cycle) induced a dopamine transient only at stimulus onset (*Figure 3C*) that was indistinguishable from the response to constant illumination (*Figure 2D*). This is in contrast to earlier repeated white LED experiments (*Figure 2—figure supplement 1*) with a long ISI (90–180 s) that showed no trial-by-trial reduction in the dopaminergic response to light across a range of LED irradiances. To further explore how LNAc dopamine responses are affected by exposure history, we measured the response to 5 µW/cm$^2$ white LED light (1 s duration × five trials with a 100 s ISI) in stimulus-naïve mice before and after three hundred consecutive 1 s light exposures (1 s ISI) during the same session (*Figure 3D*). Compared to baseline measurements, there was a significant reduction (31.6%) in the peak dLight1 response to the 5 µW/cm$^2$ light after exposure to three hundred LED exposures. This reduction in the dopaminergic response to white LED light was transient, as peak dLight1 magnitude returned to baseline when mice were re-exposed to the light stimulus 48 hr later (*Figure 3D*). Therefore, the habituation of the dopamine response to repeated light stimuli is more strongly influenced by stimulus frequency than the total number of previous exposures.

During repeated stimulus experiments, the greatest reduction in the peak LNAc dopamine response to light stimuli occurred between the first and second light pulse in each stimulus train. In order to better characterize this phenomenon, we varied the duration of the first stimulus to determine if the total amount of initial light exposure modulates the dopaminergic response to a subsequent stimulus (*Figure 3E*). We found that the dopaminergic response to a 1 s white LED test stimulus was not significantly different when preceded by either a 300 s or 1 s preconditioning light stimulus 1 s earlier (*Figure 3F*). No difference in dLight1 response to the preconditioning stimulus was observed between conditions (*Figure 3F*). We did observe, however, that the 300 s preconditioning stimulus produced a dopaminergic response at a light offset, whereas the 1 s preconditioning stimulus did not (*Figure 3G*). This observation is consistent with rebound excitation exhibited by light-adapted OFF and ON-OFF retinal ganglion cells when a prolonged light stimulus is discontinued (*Tikidji-Hamburyan et al., 2015*; *Drinnenberg et al., 2018*). Taken together, our findings indicate that ISI is a more significant determinant of stimulus-to-stimulus dopamine release habituation than light stimulus duration.

## Wavelength and photoreceptor contributions to the dopaminergic response to light

In these and previous experiments (*Robinson et al., 2019*), we employed a white LED light to induce striatal dopamine release; however, this light source is composed of multiple wavelengths throughout the visible spectrum. Therefore, we next investigated if light-evoked dopamine release exhibits wavelength specificity. This is additionally germane given the widespread use of molecular and optical technologies in rodents that require delivery of specific wavelengths of visible light in order to probe

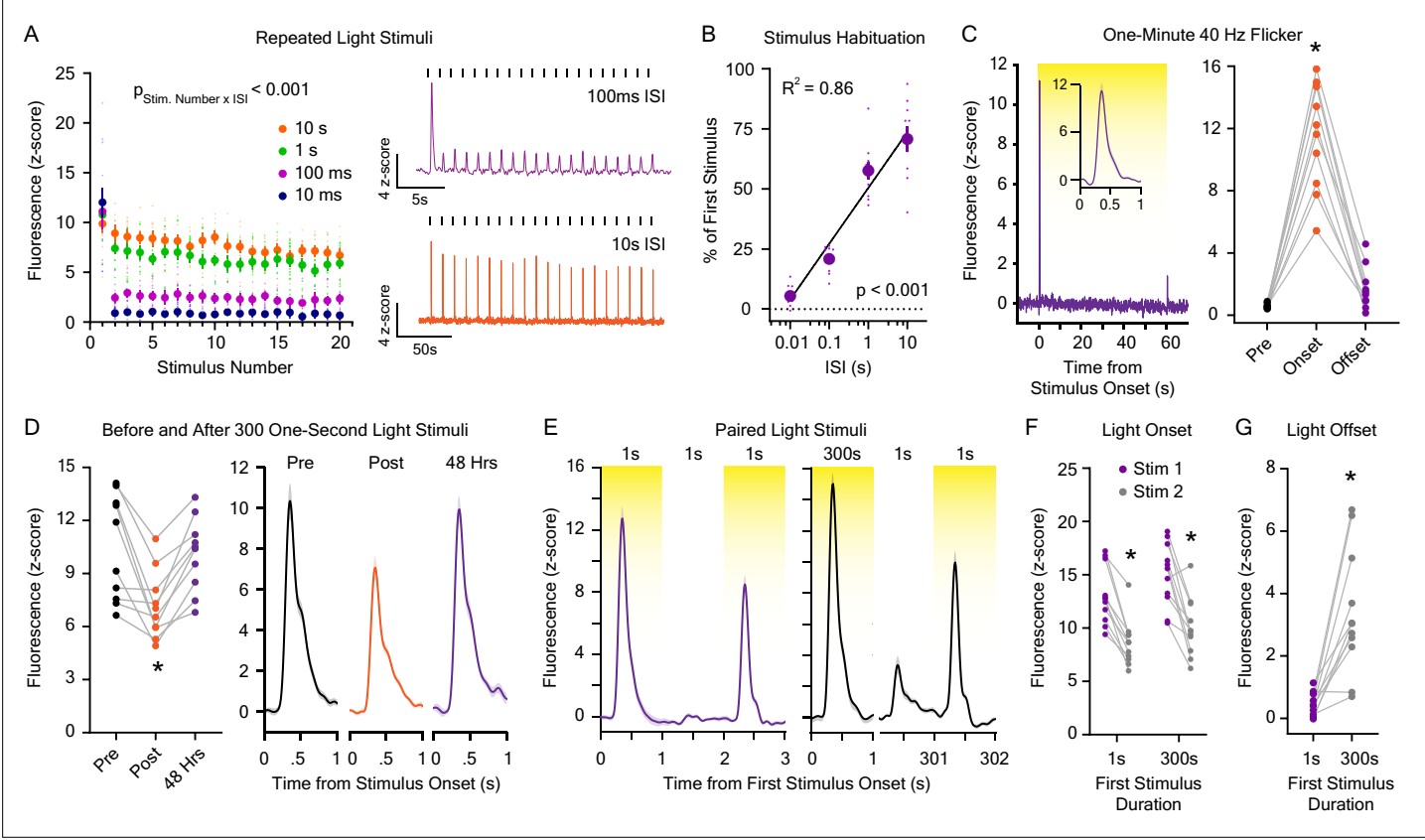

**Figure 3.** Dopaminergic responses to repeated light stimuli. (**A**) (*Left*) Dopamine release evoked by 20 1 s white light emitting diode (LED) stimuli was reduced with repeated exposures and was dependent on the interstimulus interval (ISI; 10 ms – 10 s; n=9; 2-way repeated measures ANOVA; $F_{57,456}$ = 9.54, $p_{stimulus\ number\ x\ interstimulus\ interval}$ <0.001; $F_{19,456}$ = 72.98, $p_{stimulus\ number}$ <0.001; $F_{3,456}$ = 63.91, $p_{interstimulus\ interval}$ <0.001). (*Right*) Averaged dLight1 fluorescent traces showing the dopaminergic response to 20 1 s white LED light pulses with a 100 ms ISI (*purple*) or 10 s ISI (*orange*). (**B**) Total habituation of the peak dLight1 response to repeated stimuli (shown as the peak response to the 20[th] stimulus as a percentage of the 1[st] stimulus) is dependent on the duration of the interstimulus interval (n=9; 1-way repeated measures ANOVA; $F_{3,24}$ = 104.0, p<0.001). Data are shown with a semi-log fit (y-intercept: 48.92%, slope: 22.58% $s^{-1}$, $R^2$ = 0.86). (**C**) (*Left*) Averaged dLight1 trace showing lateral NAc (LNAc) dopamine evoked by a 60 s presentation of 40 Hz white LED flicker (*inset: response during the first second after stimulus onset*) ± SEM. (*Right*) 40 Hz flicker only evoked significant dopamine release at stimulus onset (n=10; 1-way repeated measures ANOVA; $F_{2,18}$ = 100.4, p<0.001). Bonferroni *post hoc* tests confirmed that the dLight1 peak at LED onset was greater than the baseline and offset responses, which did not differ from each other (p=0.09). (**D**) (*Left*) The dLight1 response to a 1 s white LED stimulus in stimulus-naïve mice was reduced after the presentation of 300 1 s LED stimuli with a one-second ISI but returned to baseline 48 hr later (n=10; one-way repeated measures ANOVA with Bonferroni *post hoc* tests; $F_{2,18}$ = 12.4, p=0.002). (*Right*) Averaged dLight1 fluorescent traces showing the dopaminergic response to 1 s LED light pulses before (*black*) or after (*orange*) 300 1 s LED stimuli, as well as 48 hr later (*purple*). (**E**) Averaged dLight1 fluorescent traces showing the dopaminergic response to a 1 s white LED stimulus 1 s after a 1 s (*left*) or 300 s (*right*) preconditioning stimulus ± SEM. (**F**) The dLight1 response to a 1 s white LED test stimulus was not dependent on the length of the preconditioning stimulus (n=11; 2-way repeated measures ANOVA; $F_{1,10}$ = 0.27, $p_{initial\ stimulus\ length\ x\ stimulus\ Number}$ = 0.61; $F_{1,10}$ = 3.83, $p_{initial\ stimulus\ length}$ = 0.08; $F_{1,10}$ = 55.10, $p_{stimulus\ number}$ <0.001). Bonferroni *post hoc* tests revealed that the dLight1 response to the test stimulus onset was significantly smaller than the response to the onset of the preconditioning stimulus, regardless of its duration. There was no difference between the dLight1 response to the onset of the preconditioning (p=0.11) or test stimulus (p=0.40) between experiments. (**G**) The dLight1 response to light offset was larger for a 300 s light stimulus compared to a 1 s light stimulus (n=11; paired *t*-test; $t_{10}$=4.91, p<0.001). In all panels, * indicates p<0.05.

The online version of this article includes the following source data for figure 3:

**Source data 1.** Source data and associated statistical testing results for *Figure 3*.

neural activity, structure, or biology (*Fenno et al., 2011*; *Resendez and Stuber, 2015*; *Sabatini and Tian, 2020*). In order to determine if the dLight1 response varied by wavelength, we measured dopamine release induced by 10 s exposures to environmental ultraviolet (UV; 360 nm), blue (475 nm), green (555 nm), red (635 nm), and far-red (730 nm) light across a 100,000-fold range of irradiances (1 nW/cm² to 100 µW/cm²). These experiments revealed the broad sensitivity of the mesolimbic dopamine system to light across the visual spectrum (*Figure 4A–B*). The dopamine response was least

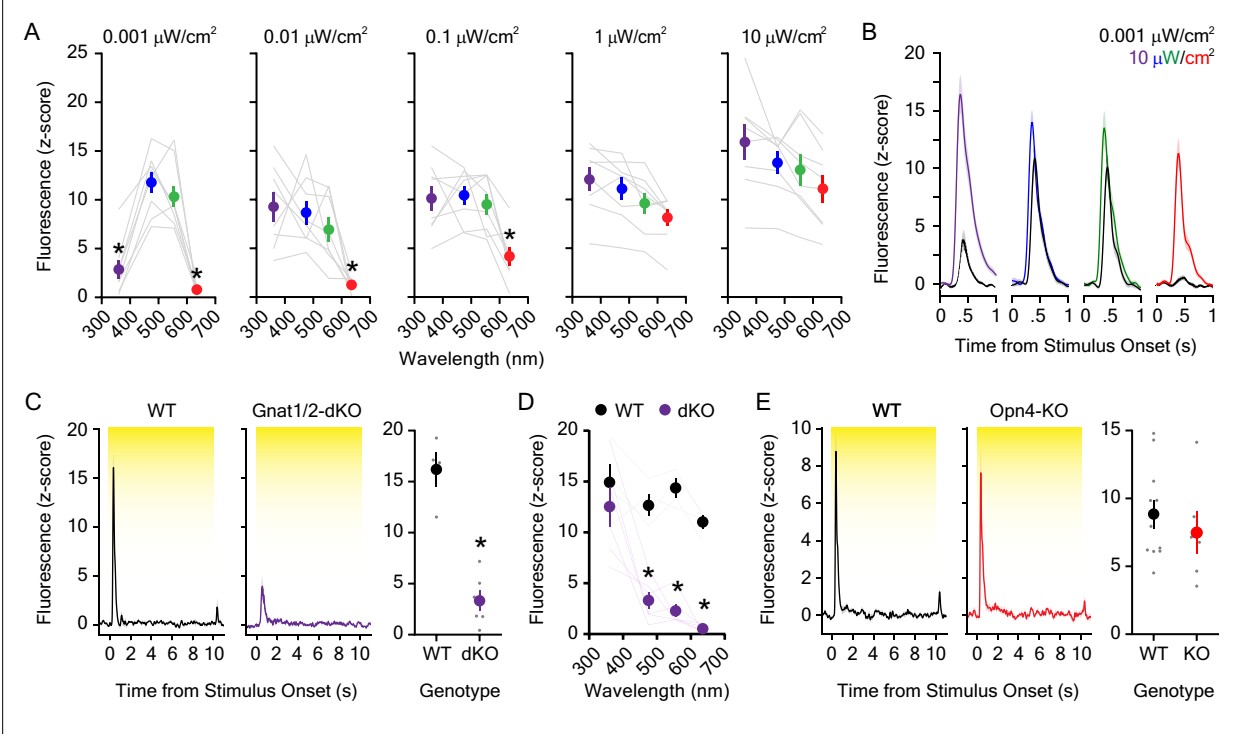

**Figure 4.** Dopaminergic responses to individual wavelengths across the visual spectrum. (**A**) The dopaminergic response to UV (360 nm), blue (475 nm), green (555 nm), and red (635 nm) light emitting diode (LED) light was wavelength and irradiance-dependent (n=8; 2-way repeated measures ANOVA; $F_{12,84} = 9.63$, $p_{wavelength \times irradiance}$ <0.001; $F_{3,84} = 37.59$, $p_{wavelength}$ <0.001; $F_{4,84} = 10.08$, $p_{irradiance} = 0.004$). Bonferroni *post hoc* tests revealed that dopamine evoked by UV and the red light was smaller than blue and green wavelengths at the lowest irradiance tested (0.001 µW/cm²). The dLight1 response to the red LED was also significantly lower than blue and green LEDs at irradiances of 0.01 µW/cm² and 0.1 µW/cm². For comprehensive reporting of all significant *post hoc* tests across irradiances and wavelengths, see the full statistical testing results in the source data file that accompanies this figure. (**B**) Averaged dLight1 trace showing lateral NAc (LNAc) dopamine evoked by either 0.001 µW/cm² (1 nW/cm²) or 10 µW/cm² UV, blue, green, or red LEDs ± SEM. (**C**) The dopaminergic response to 5.0 µW/cm² white light was significantly reduced in *Gnat1/2* double knockout (dKO) mice relative to wildtype controls ($n_{WT}$ = 4, $n_{KO}$ = 6; unpaired *t*-test; $t_8$=7.08, <0.001). (**D**) The reduction in the dLight1 response to 10 µW/cm² light in Gnat1/2 dKO was wavelength dependent (two-way repeated measures ANOVA; $F_{3,24}$ = 7.02, $p_{genotype \times wavelength}$ = 0.002; $F_{3,24}$ = 17.54, $p_{wavelength}$ = 0.003; $F_{1,24}$ = 85.80, $p_{genotype}$ <0.001). Bonferroni *post hoc* tests revealed that the dLight1 response to blue (475 nm), green (555 nm), and red (635 nm) light was lower in Gnat1/2 mice relative to wildtype littermates. (**E**) The dopaminergic response to 5.0 µW/cm² white light was not different in *Opn4* (melanopsin) knockout mice relative to wildtype controls ($n_{WT}$ = 11, $n_{KO}$ = 6; unpaired *t*-test; $t_{15}$=0.75, p=0.46).

The online version of this article includes the following source data and figure supplement(s) for figure 4:

**Source data 1.** Source data and associated statistical testing results for *Figure 4*.

**Figure supplement 1.** Dopamine responses to invidual light wavelengths in wildtype, *Opn4* knockout, and *Gnat1/2* double knockout mice.

**Figure supplement 1—source data 1.** Source data and associated statistical testing results for *Figure 4—figure supplement 1*.

sensitive to UV and red light when the irradiance was low (1 nW/cm²; *Figure 4A–B*), and far-red light (730 nm) only induced dopamine release when the irradiance was high (100 µW/cm²; *Figure 4—figure supplement 1*). The ability of red light to induce dopamine release at intensities as low as 0.1 µW/cm² is consistent with research that rodents are better at perceiving red wavelengths than is commonly acknowledged (*Danskin et al., 2015*; *Nikbakht and Diamond, 2021*; *Vinberg et al., 2019*). Whereas the dLight1 response to UV and the red light was irradiance-dependent, the response to blue and green light remained robust across the entire irradiance range (*Figure 4A*). These experiments indicate that the mesolimbic dopamine system is responsive to all visible wavelengths yet is most sensitive to blue and green light.

The mouse visual system utilizes numerous opsin proteins for image-forming and non-imaging forming phototransduction with unique wavelength sensitivities. These include the rod opsin rhodopsin for scotopic vision ($\lambda$ max ~500 nm) and short ($\lambda_{max}$ ~360 nm) and medium/long wavelength ($\lambda_{max}$ ~508 nm) cone opsins for photopic vision. Additionally, melanopsin ($\lambda_{max}$ ~480 nm) is expressed in intrinsically photosensitive retinal ganglion cells (ipRGCs) that mediate circadian entrainment, the

pupillary light reflex, and light-regulated changes in mood (*Panda et al., 2003*; *Hattar et al., 2003*; *Fernandez et al., 2018*). While it has been hypothesized that ipRGCs engage VTA dopamine neurons via hypothalamic intermediates (*Zhang et al., 2021*), the role of melanopsin in the dopaminergic response to light is unknown. In order to parse the role of visual opsins versus melanopsin in the meso-limbic response to dark-to-light transitions, we performed LNAc dLight1 recordings in *Opn4* (melan-opsin) knockout mice and *Gnat1/Gnat2* double knockout mice (*Gnat1/2*-dKO). *Gnat1 and 2* knockout mice lack expression of rod and cone α-transducin, respectively, and exhibit loss of signal transduction through these photoreceptors (*Deng et al., 2009*; *Yao et al., 2018*). Compared with wildtype litter-mates, *Gnat1/2*-dKO mice displayed a robust reduction in the dopaminergic response to a 5 µW/cm$^2$ white LED (*Figure 4C*) and an increase in the dLight1 response latency (*Figure 4—figure supplement 1*). Light-evoked dopamine release was not abolished, however, in these mice (*Figure 4C*). Spectral analysis indicated that *Gnat1/2*-dKO mice retain sensitivity to UV light (*Figure 4D*, *Figure 4—figure supplement 1*), which may be indicative of residual cone-based vision (*Allen et al., 2010*). Conversely, loss of melanopsin expression in *Opn4* knockout mice (*Panda et al., 2002*) did not affect the dLight1 response to the white light stimulus (*Figure 4E*, *Figure 4—figure supplement 1*). These findings indi-cate that light-evoked dopamine release is rod and cone-dependent and may not involve melanopsin.

## Discussion

In these investigations, we used the genetically-encoded dopamine sensor dLight1 to demonstrate that lateral NAc dopamine release can encode rapid changes in luminance but not looming threat intensity. We found that rapid dark-to-light transitions evoked time-locked dopamine responses at stimulus onset at irradiances as low as 0.2 nW/cm$^2$, which is in line with findings that mice see over a 100 million-fold range of light intensity beginning at ~4 µcd/m$^2$ (*Umino et al., 2008*). The magnitude of these dopaminergic responses was highly dependent on light stimulus frequency and transition rate rather than duration or novelty. In fact, high amplitude LNAc dLight1 responses to a white LED persisted after hundreds of exposures. Although mesolimbic dopamine systems regulate wakefulness (*Eban-Rothschild et al., 2016*) and exhibit circadian oscillation (*Korshunov et al., 2017*), the time of testing did not appear to be a significant contributor to our findings. Sudden dark-to-light transitions are highly salient to nocturnal rodents that must avoid detection by visual predators (*Thompson et al., 2010*), so it is possible that the dopaminergic response to light represents a specialized saliency signal that helps the animal alert to stimuli that require motivated responses to promote survival (*Schultz and Romo, 1990*; *Horvitz, 2000*).

In primates, midbrain dopaminergic neurons fire in response to unrewarded visual stimuli inde-pendent of physical salience or novelty (*Kobayashi and Schultz, 2014*). However, the magnitude of these responses is significantly more robust in rewarded environments, suggesting that dopaminergic responses to sensory stimuli primarily serve to promote interaction with potentially rewarding objects (*Kobayashi and Schultz, 2014*) or enable novel action discovery that could lead to reward (*Redgrave and Gurney, 2006*). While there was no conditioned reward explicitly available in the present study, dopaminergic responses to visual stimuli may relate to changes in environmental context that predict future reward availability based on past experiences, such as rapid lighting changes that occur when mouse cages are opened to provide food pellets. Additionally, dopamine release in the medial NAc core in response to unconditioned auditory cues has been shown to facilitate latent inhibition, in which habituation to repeated presentation of a neutral sensory stimulus reduces its ability to drive cue-outcome learning (*Kutlu et al., 2022*). These considerations emphasize the need for future studies to firmly establish the ethological and neurobiological importance of dopaminergic responses to envi-ronmental light, especially when the relationship to a past or future reward is not obvious, as in our experiments.

The LNAc dopamine response to light may also be influenced by the ability of dopaminergic neurons to signal sensory prediction errors (*Takahashi et al., 2017*; *Howard and Kahnt, 2018*; *Stal-naker et al., 2019*). In an elegant set of experiments in rats, VTA dopaminergic neurons were shown to fire in response to unexpected changes in the sensory properties of a reinforcer when the relative subjective value was unchanged (e.g. switching the flavor of an equally palatable Kool-Aid reward) (*Takahashi et al., 2017*; *Stalnaker et al., 2019*). Thus, it is possible that the ISI-dependent attenuation of the dopamine response to repeated light stimuli that we observed is not true habituation. Rather, the dopaminergic response to light response may have decayed rapidly because events occurring

closer in time are inherently more predictable, thus reducing the sensory prediction error and evoked dopamine release. While this hypothesis could be tested by unexpectedly altering the stimulus wavelength to induce a sensory prediction error during a train of repeated light exposures, one would need to carefully account for the spectral overlap of mouse visual opsins and the non-uniform distribution of cone opsins across the retina to ensure that changes in dopamine release were due to changes in chromaticity rather than luminance, which would confound the results. The use of transgenic mice genetically engineered to express the red human cone opsin ($\lambda_{max}$ = 556 nm) in place of the mouse M-opsin (*Smallwood et al., 2003*; *Lall et al., 2010*) may be useful in future efforts to test this hypothesis, as it would allow for activation of cones with isoluminant blue and red light independent of melanopsin (*Brown et al., 2010b*; *Allen et al., 2011*).

In these studies, we demonstrate that LNAc dopamine is broadly evoked by wavelengths across the visual spectrum. Given the high proportion of rods in the mouse retina (~97% of photoreceptors) (*Jeon et al., 1998*) and the reduced sensitivity of dopaminergic responses to 360 and 635 nm light at lower irradiances, it is probable that rod-based phototransduction is primarily responsible for visually-evoked dopamine release under dim (scotopic) lighting conditions. Conversely, rod and cone opsins likely contributed to dLight1 signals in the photopic range. These hypotheses are supported by our observation that genetic disruption of rod and cone-based signaling in *Gnat1/2*-dKO mice substantially attenuated the dopaminergic response to light. *Gnat1/2*-dKO mice retained sensitivity to high irradiance UV light, which was most likely caused by incomplete loss of cone-based vision in this model (*Allen et al., 2010*). We cannot, however, rule out the involvement of UV-sensitive non-visual opsins in our observed findings, such as neuropsin (*Opn5*), which is maximally activated by 380 nm light (*Tarttelin et al., 2003*). Neuropsin-expressing retinal ganglion cells project to multiple limbic regions (*Sasaki et al., 2021*), and this opsin promotes thermogenesis via intrinsically light-sensitive glutamatergic neurons in the preoptic area (*Zhang et al., 2020*). While melanopsin-expressing ipRGCs are hypothesized to engage VTA outputs via a disynaptic circuit involving the preoptic area (*Zhang et al., 2021*), we found that *Opn4* knockout had no effect on the ability of light to evoke LNAc dopamine. Given that ipRGCs receive rod and cone input via the retinal synaptic network (*Güler et al., 2008*; *Lall et al., 2010*; *Altimus et al., 2010*), it is possible that these neurons contribute to light-evoked dopamine release independent of melanopsin. Thus, functional lesioning studies will be required to elucidate the role of non-image forming visual pathways in the dopaminergic encoding of visual stimuli.

Visual information is conveyed from the retina to the brain via the axons of retinal ganglion cells that synapse in downstream nuclei to mediate image processing, circadian entrainment, pupillary reflexes, gaze orientation, etc. (*Peirson et al., 2018*). While thalamocortical visual pathways are required for conscious visual perception, neither the primary visual cortex (V1) nor the visual thalamus (e.g. lateral geniculate nucleus) significantly innervates ventral midbrain dopamine neurons (*Watabe-Uchida et al., 2012*). Previous work by Redgrave and colleagues suggest that dopaminergic responses to light are driven by the superior colliculus (SC) (*Comoli et al., 2003*; *Dommett et al., 2005*; *Takakuwa et al., 2017*), which receives direct input from retinal ganglion cells (*Dhande and Huberman, 2014*) in its superficial layers and promotes motivated behavior via deep motor-output layers (*Branco and Redgrave, 2020*). SC glutamatergic projection neurons directly synapse onto VTA (*Solié et al., 2022*) and substantia nigra pars compacta dopamine neurons (*Huang et al., 2021*), both of which project to the lateral NAc (*Beier et al., 2015*; *Poulin et al., 2018*). Likewise, optogenetic stimulation of SC neuron somata is sufficient to evoke lateral NAc dopamine release *in vivo* (*Robinson et al., 2019*). While these observations support a role for the SC in dopaminergic responses to light, the relative contribution of different visual processing centers to our findings is an important area of future study.

One important question not addressed by the current study is whether the dopaminergic response to unconditioned visual stimuli is consistent across striatal sub-regions or shows regional heterogeneity. We performed dLight1 recordings in the lateral NAc, which receives dopaminergic innervation from the lateral VTA and medial substantia nigra pars compacta (*Yang et al., 2018*; *Farassat et al., 2019*). LNAc-projecting VTA dopamine neurons have broad axonal arbors covering the dorsal striatum, olfactory tubercle, and NAc core (*Beier et al., 2015*), which may indicate that dopamine release encodes visual stimulus properties across these sub-regions. Conversely, the NAc medial shell is innervated by more medially located VTA dopaminergic neurons (*Lammel et al., 2011*; *Beier et al., 2015*) whose axonal arbors are primarily restricted to this downstream site (*Beier et al., 2015*). These differences in connectivity may explain previously observed variations in dopaminergic encoding across

NAc subregions. For example, VTA axon terminals and dopamine release in the LNAc encode both stimulus valence and prediction errors (*de Jong et al., 2019*; *Robinson et al., 2019*; *Yuan et al., 2019*), similar to responses in the adjacent NAc core (*de Jong et al., 2019*; *Patriarchi et al., 2018*). This is in contrast to the NAc medial shell, where dopaminergic axons are strongly activated by appetitive and aversive motivational stimuli but not reward predictive cues (*de Jong et al., 2019*). At this time, how and if NAc medial shell dopamine encodes visual stimulus characteristics is unknown and represents an important future direction for study.

Mesolimbic dopaminergic circuits are thought to play a role in the pathophysiology of several neuropsychiatric conditions, including disorders of impulse control, schizophrenia, and neurodevelopmental disorders (*Li et al., 2006*; *Purper-Ouakil et al., 2011*; *Maia and Frank, 2017*; *Robinson and Gradinaru, 2018*), including NF1 (*Brown et al., 2010a*; *Diggs-Andrews et al., 2013*; *Anastasaki et al., 2015*). Patients with NF1 exhibit high rates of attention-deficit/hyperactivity disorder (*Mautner et al., 2015*; *Miguel et al., 2015*), in which difficulties with attentional orientation are associated with a diminished ability to suppress distractive stimuli (*Aboitiz et al., 2014*) such that irrelevant environmental cues are assigned exaggerated stimulus salience (*Tegelbeckers et al., 2015*). Previously in eLife, we showed that dopaminergic responses to light are enhanced in NF1 model mice and correlate with disruptions in the expression of conditioned behavior (*Robinson et al., 2019*). Our current findings suggest that these responses reflect changes in the encoding of environmental lighting conditions and, given their correlation with phenotypic expression, may reflect altered stimulus saliency. Aberrant sensory processing and motivational dysregulation are common features of neurodevelopmental disorders, including syndromic and non-syndromic forms of autism spectrum disorder (*Behrmann et al., 2006*; *Tomchek and Dunn, 2007*; *Robinson and Gradinaru, 2018*). Therefore, better characterization of the functional interplay between visual processing and dopaminergic circuitry may improve our pathophysiological understanding of these disorders.

# Materials and methods

## Key resources table

| Reagent type (species) or resource | Designation | Source or reference | Identifiers | Additional information |
|---|---|---|---|---|
| Recombinant DNA reagent | pAAV-hSyn-dLight1.2 | Addgene | Cat#: 111068 RRID:Addgene_111068 | Produced by Addgene in the AAV5 serotype |
| Software, Algorithm | Python 3.8 | Python Software Foundation | RRID:SCR_008394 | |
| Software, Algorithm | Fiber Photometry Trace Processing | Tucker-Davis Technologies | | https://www.tdt.com/docs/sdk/offline-data-analysis/offline-data-python/examples/FibPhoEpocAveraging/ |
| Software, Algorithm | Bonsai 2.6.3 | Bonsai Foundation CIC | RRID:SCR_017218 | |
| Software, Algorithm | Looming Visual Stimulus Generation | Austen Fisher, Robinson Lab | | https://github.com/jelliottrobinson/BonsaiLoomStim |
| Software, Algorithm | ABET II Software for Operant Control | Lafayette Instrument Company | Model 89501 | |
| Software, Algorithm | Ethovision XT 17 | Noldus Information Technology | RRID:SCR_000441 | |
| Software, Algorithm | GraphPad Prism 9 | GraphPad Software, Inc | RRID:SCR_002798 | |
| Software, Algorithm | Data Science Workbench 14.0.0.15 | TIBCO Software, Inc | RRID:SCR_014213 | |
| Other | Mono Fiber-Optic Cannula | Doric Lenses, Inc | Cat#: MFC_400/430–0.66_6 mm_MF1.25_FLT | OD: 400 μm, Length: 6 mm |
| Other | Mono Fiber-Optic Patch Cable | Doric Lenses, Inc | Cat#: MFP_400/430/1100–0.57_1 m_FCM-MF1.25_LAF, Doric Lenses Inc | OD: 400 μm, Length: 1 m |

## Experimental animals

Experimental subjects were adult male and female C57Bl/6J mice (the Jackson Laboratory Stock No: 000664), homozygous *Opn4* knockout mice (*Panda et al., 2002*), or homozygous *Gnat1/2* knockout mice (*Gnat1*[-/-], *Gnat2*[cpfl-3] mice; the Jackson Laboratory Stock No: 033163) that were greater than 12 weeks of age. Animals were paired or group housed (3–4 per group) throughout the duration of the experiment in a vivarium on a 14 hour/10 hr light/dark cycle (lights on at 0600 hr, lights off at 2000 hr) with *ad libitum* access to food and water. All experiments were performed during the light phase of the vivarium light/dark cycle, except when white LED exposure was performed 2–3 hr into the dark phase, as shown in *Figure 2G*. Animal husbandry and experimental procedures involving animal subjects were conducted in compliance with the Guide for the Care and Use of Laboratory Animals of the National Institutes of Health and approved by the Institutional Animal Care and Use Committee (IACUC) and by the Department of Veterinary Services at Cincinnati Children's Hospital Medical Center (CCHMC) under IACUC protocol 2020–0058. Mice were excluded from studies if they could not complete an entire experiment due to loss of the brain implant or if there was no dynamic photometry signal six weeks after surgery. Following the completion of experiments, mice were transcardially perfused with 4% paraformaldehyde in phosphate-buffered saline so that the photometry fiber location could be determined histologically.

## Surgical procedures

Stereotaxic viral vector injections and optical fiber implantation surgeries for dLight1 were performed as previously described (*Robinson et al., 2019*). This procedure was similar to the published protocol of Tian and colleagues (*Patriarchi et al., 2019*). In brief, mice were anesthetized with isoflurane (1–3% in 95% $O_2$/5% $CO_2$ provided via nose cone at 1 L/min), the scalp was shaved and sterilized with chlorhexidine surgical scrub, the skull surface was exposed, and a craniotomy hole was drilled over the lateral NAc (antero-posterior: 1.2 mm, medio-lateral: 1.6 mm relative to Bregma). 800–1000 nL of a AAV5-hSyn-dLight1.2 vector (~1 × 10$^{13}$ viral genomes/mL, obtained from Addgene; catalog #AAV5-111068) was delivered into the LNAc (antero-posterior: 1.2 mm, medio-lateral: 1.6 mm, dorso-ventral: –4.2 mm relative to Bregma) using a blunt or beveled 34 or 35-gauge microinjection needle within a 10 uL microsyringe (NanoFil, World Precision Instruments) controlled by a microsyringe pump with SMARTouch Controller (UMP3T-1, World Precision Instruments) over 10 min. Following viral injection, a 6 mm long, 400 µm outer diameter mono fiber-optic cannula (MFC_400/430–0.66_6 mm_MF1.25_FLT, Doric Lenses Inc) with a metal ferrule was lowered to the same stereotaxic coordinates and affixed to the skull surface with C&B Metabond (Parkel Inc) and dental cement. Mice were given 5 mg/kg carprofen (s.c.) intraoperatively and for two days postoperatively for pain. Mice were allowed a minimum of five weeks for surgical recovery and virus expression prior to participation in behavioral studies.

## Fiber photometry

Fluorescent signals were monitored using an RZ10x fiber photometry system from Tucker-Davis Technologies, which allowed for dLight1 excitation and emission light to be delivered and collected via the same implanted optical fiber. Our system employed a 465 nm LED for sensor excitation and a 405 nm LED for isosbestic excitation. Light was filtered and collimated using a six-channel fluorescent MiniCube [FMC6_IE(400-410)_E1(460–490)_F1(500–540)_E2(555–570)_F2(580–680)_S] from Doric Lenses, Inc, which was coupled to the implanted optical fiber via a one-meter, low autofluorescence fiber optic patch cable (MFP_400/430/1100–0.57_1_FCM-MF1.25LAF, Doric Lenses Inc). The emission signal from 405 nm isosbestic excitation was used as a reference signal to account for motion artifacts and photo-bleaching. A first-order polynomial fit was applied to align the 465 nm signal to the 405 nm signal. Then, the polynomial fitted model was subtracted from the 465 nm channel to calculate ΔF values. The code for performing this function was provided by Tucker-Davis Technologies, Dr. David Root (University of Colorado, Boulder), and Dr. Marisela Morales (NIDA); it is available at: https://www.tdt.com/docs/sdk/offline-data-analysis/offline-data-python/examples/FibPhoEpocAveraging/ (*Root et al., 2022*).

During behavioral experiments, the ΔF time-series trace was z-scored within epochs to account for data variability across animals and sessions, as described by Morales and colleagues (*Barker et al., 2017*). When fiber photometry was performed during sensory stimulus exposure experiments,

dLight1 signals were synchronized to stimulus onset via the delivery of TTL pulses to the photometry system. Generally, we tried to design experiments where photometry signals could be averaged across repeated trials to limit background noise. Peak data (magnitude, latency, and full width at half-maximal intensity) was analyzed using Python.

## Visual stimulus exposure

Visual stimuli were delivered to mice during fiber photometry recordings with unique stimuli presented to the same subject during different experimental sessions. Experiments in C57Bl/6 J mice were performed in the following order with a minimum of 48 hr between experiments: overhead disc exposures, full field fades, white LED light stimuli, tone exposures, and individual wavelength exposures. Experiments in *Figure 3D* were performed in a separate cohort of stimulus-naïve mice. Experiments in *Gnat1/2* knockout and wildtype littermates were performed in the following order: white LED light stimuli, and individual wavelength exposures. Opn4 mice underwent white LED exposure only. For overhead disc stimuli, photometry was performed within a custom setup that featured a 24-inch LCD mounted 25.4 cm above mouse level in a light and sound attenuating chamber (Model 83018DDP, Lafayette Instrument Company). Stimuli (looming, static, and receding discs; full screen fades; etc.) were generated on the LCD display using Bonsai (*Lopes et al., 2015*), which also controlled delivery of a TTL pulse to the photometry system via a BNC cable to timestamp stimulus onset. The TTL pulse was generated with an Arduino Uno Rev3 microcontroller. During each experiment, mice were placed within the bottom of a clean shoebox cage with a thin layer cob bedding in the light and sound-attenuating chamber underneath the LCD. Looming discs expanded from 0 cm to 30.5 cm over 0.84 s and froze at full expansion for 0.26 s, encompassing 61.9 degrees of visual angle, as previously described (*Evans et al., 2018*; *Yilmaz and Meister, 2013*). Receding discs shrunk from 30.5 cm to 0 cm over 0.84 s. Static discs maintained their 30.5 cm diameter throughout the duration of the stimulus. During each stimulus train, five discs were shown consecutively with a 0.5 s interstimulus interval (ISI). Mice were exposed to five stimulus trains with a 600 s inter-trial interval (ITI) on each experimental day. In a separate experiment, single full field fades from black to light gray (0–10 s fade duration) was delivered via the LCD screen across five trials with a 120 s ITI on different days of testing.

White LED exposures were delivered via the house light of a modular conditioning chamber (Model 80015NS, Lafayette Instruments Company) placed within the light and sound attenuating box and controlled by ABET II software (Lafayette Instrument Company). A TTL breakout adapter (Model 81510) was used to synchronize stimulus delivery with the photometry recording. Single 10 s light stimuli were delivered across ten trials with a randomized ITI between 90 and 180 s. Glass neutral density filters were used to attenuate the irradiance when necessary (0.1–3.0 OD, HOYA Filter USA and/or Edmund Optics TECHSPEC filters). Because ND filters could not be changed mid-testing session, responses to each light intensity were recorded on different testing days. Trains of twent 1 s light stimuli with variable ISIs (10 ms – 10 s) were delivered across five trials (100 total exposures) with a 300 s ITI on different testing days. Five 1 s light stimuli with a 100 s ISI were delivered before and after 300 1 s light stimuli with a one-second ISI; 100 s separated the 300 1 s stimuli and each 100 s ISI stimulus train. The five-stimulus train with a 100 s ISI was repeated 48 hr later in the same group of mice. 1 min of 40 Hz flicker exposure (50% duty cycle) was repeated across 5 trials with a 120 s ITI. For paired light stimuli experiments, a 1 s white LED stimulus was delivered 1 s after a 300 s or 1 s light stimulus across ten trials (five trials/stimulus pair presented in a random order) during the same testing session. Each trial was separated by 300 s.

Individual wavelength light stimuli were generated with a Lumencor Aura III LED light engine, which was triggered via TTL inputs from the Lafayette Instruments TTL breakout adapter and controlled by ABET II. The liquid light guide that delivered the visual stimulus was positioned in the approximate location of the white LED within the testing chamber. LED light power (measured at mouse level with a Thor Labs PM100D optical power meter with S130VC photodiode sensor) was modulated using the onboard Lumencor graphical user interface and, when necessary, attenuated via the use of glass neutral density filters (0.1–3.0 OD, HOYA Filter USA and/or Edmund Optics TECHSPEC filters) placed in front of the liquid light guide outlet within a custom housing. Ten-second single-wavelength stimuli were delivered in random order with a randomized ITI (140–200 s) to achieve five total exposures per color per mouse. Because LED power could not be adjusted mid-testing session, responses to different irradiances were measured on separate testing days.

## Auditory stimulus exposure

Auditory stimulus exposures were performed in the modular testing chamber within the light and sound attenuating enclosure similar to single white LED exposures. A 10 s 80 dB tone (1–16 kHz; generated via Lafayette Instruments 7 Tone Generator Model 81415 M) was presented via a speaker (0.25–16 kHz; Model 80135 M14, Lafayette Instrument Company) across five trials with a randomized ITI (140–200 s) during the same experimental session.

## Looming stimulus assay

The looming stimulus assay was performed as previously described (*Yilmaz and Meister, 2013*) using an apparatus built to the specifications of *Evans et al., 2018*. The apparatus featured a 20.3 cm (w) × 61 cm (l) × 40.6 cm (h) clear, open, rectangular acrylic arena with a dark, infrared (IR) light-transmitting shelter at one end and a 'threat zone' at the opposite end that housed a 9 cm clear plastic petri dish to encourage exploration outside of the shelter. A 15.6-inch monitor was mounted above the arena so that discs (19.5 cm maximum diameter encompassing 27 degrees of visual angle) could be presented to the mice when they entered the threat zone. The arena floor was backlit with an infrared light (880 nm back-lit collimated backlight, Advanced Illumination) to improve mouse tracking under dim light conditions. The entire apparatus was placed inside a custom light-attenuating enclosure for testing. During testing, mice were recorded with a Basler acA2040-120 um camera with an Edmunds Optics TECHSPEC 6 mm C Series fixed focal length lens, and real-time position tracking was performed with Bonsai. This allowed for the presentation of the overhead looming, receding, or static disc stimulus to be automatically triggered when the animal was in the threat zone following a 10 min habituation period. Mouse position and velocity data were analyzed *post hoc* using Ethovision XT software (Noldus Information Technology) and Python. Note: In *Video 2*, the clear, circular pedestals that separated the infrared backlight from the apparatus base can be seen with the IR camera; they were below the arena floor and inaccessible to the mouse. The setup was modified for spotlight experiments so that the pedestals would not be visible in the captured videos. Spotlight experiments were performed in the same apparatus using the same procedure described above except that a high-intensity white LED (5 mW/cm$^2$ measured at mouse level) positioned to illuminate the threat zone replaced the LCD monitor.

## Statistical analysis

Statistical analysis was performed using Python, GraphPad Prism 9 (GraphPad Software, Inc), and/or Data Science Workbench 14 (for 3-way repeated measures ANOVA; TIBCO Software, Inc). All statistical tests performed on data presented in the manuscript are stated in the figure captions and provided in detail with the corresponding source data files. For each experiment, statistical tests were chosen based on the structure of the experiment and the data set. No outliers were removed during statistical analysis. Parametric tests were used throughout the manuscript. Sample size estimates were based on studies by *Robinson et al., 2019* and power analysis performed using the sampsizepwr function in Matlab (MathWorks). When analysis of variance (ANOVA; 1-way, 2-way, 3-way, and/or repeated measures) was performed, multiple comparisons were corrected using the Bonferroni correction. When repeated measures ANOVA could not be performed due to missing values (*Figure 2—figure supplement 1C*), data were analyzed by fitting a mixed model in GraphPad Prism 9; this approach uses a compound symmetry covariance matrix and is fit using restricted maximum likelihood (REML). When results were compared to a pre-stimulus baseline, this value was defined as the amplitude of the dLight1 peak that occurred 500 ms prior to stimulus delivery. When results were compared to a 'null' stimulus, the value was defined as the dLight1 peak that occurred at the onset of a TTL that timestamped a trial in which no stimulus was delivered.

## Data and materials availability

Viral vector plasmids used in this study are available on Addgene. Codes used for fiber photometry signal extraction and analysis are available at https://www.tdt.com/docs/sdk/offline-data-analysis/offline-data-python/examples/FibPhoEpocAveraging/. Codes used for visual stimulus generation are available at https://github.com/jelliottrobinson/BonsaiLoomStim (*Gonzalez, 2023*; copy archived at swh:1:rev:8353dc51dfffd013160b14ed75fd5ae040144245). Source data is provided with each figure.

## Acknowledgements

We would like to acknowledge Dr. Ronald Waclaw and Ms. Mary Claire Casper at CCHMC for assistance with histological sample preparation and microscopy. We would also like to thank Dr. Gregory Schwartz at Northwestern University Feinberg School of Medicine and Dr. Diego Fernandez at the National Institute of Mental Health for helpful discussions regarding technical considerations and/or interpretation of the experimental findings. This work was funded by a Cincinnati Children's Research Foundation Trustee Award, a Simons Foundation Autism Research Initiative (SFARI) Bridge to Independence Award (663007), a SFARI Supplement to Enhance Equity and Diversity (SEED) Award, and a Gilbert Family Foundation Neurofibromatosis Gene Therapy Initiative Team Science Award to JER. LSG was supported by a National Institutes of Health Training Grant (T32 NS007453).

## Additional information

### Funding

| Funder | Grant reference number | Author |
| --- | --- | --- |
| Simons Foundation Autism Research Initiative | BTI Award 663007 | J Elliott Robinson |
| Gilbert Family Foundation | Team Science Award | J Elliott Robinson |
| Cincinnati Children's Research Foundation | Trustee Award | J Elliott Robinson |
| Simons Foundation Autism Research Initiative | SEED award | Evelin M Cotella J Elliott Robinson |
| National Institutes of Health | Training Grant T32 NS007453 | L Sofia Gonzalez |

The funders had no role in study design, data collection and interpretation, or the decision to submit the work for publication.

### Author contributions

L Sofia Gonzalez, Conceptualization, Data curation, Formal analysis, Investigation, Visualization, Writing – original draft, Writing – review and editing; Austen A Fisher, Conceptualization, Data curation, Software, Formal analysis, Investigation, Visualization, Methodology, Writing – original draft, Writing – review and editing; Shane P D'Souza, Richard A Lang, Resources, Writing – review and editing; Evelin M Cotella, Formal analysis, Writing – review and editing; J Elliott Robinson, Conceptualization, Data curation, Formal analysis, Supervision, Funding acquisition, Visualization, Methodology, Writing – original draft, Project administration, Writing – review and editing

### Author ORCIDs

Shane P D'Souza ⓘ http://orcid.org/0000-0001-6344-1434
J Elliott Robinson ⓘ http://orcid.org/0000-0001-9417-3938

### Ethics

Animal husbandry and experimental procedures involving animal subjects were conducted in compliance with the Guide for the Care and Use of Laboratory Animals of the National Institutes of Health and approved by the Institutional Animal Care and Use Committee (IACUC) and by the Department of Veterinary Services at Cincinnati Children's Hospital Medical Center (CCHMC) under IACUC protocol 2020-0058.

### Decision letter and Author response

Decision letter https://doi.org/10.7554/eLife.85064.sa1
Author response https://doi.org/10.7554/eLife.85064.sa2

## Additional files

### Supplementary files
• MDAR checklist

### Data availability
Viral vector plasmids used in this study are available on Addgene. Codes used for fiber photometry signal extraction and analysis are available at https://www.tdt.com/docs/sdk/offline-data-analysis/offline-data-python/examples/FibPhoEpocAveraging/ (*Root et al., 2022*). Codes used for visual stimulus generation are available at https://github.com/jelliottrobinson/BonsaiLoomStim (copy archived at *Gonzalez, 2023*). Source data is available in the source data files attached to the figures.

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
