## [Editor Report]

In this manuscript, Gonzalez et al. investigated the dynamics of dopamine signals in the lateral shell of the nucleus accumbens (LNAc) in response to different types of carefully defined visual stimuli. Contrary to reigning theories of dopamine signaling, the authors presented convincing evidence that LNAcc dopamine transients tracked visual sensory transitions rather than any immediately apparent motivational variable. These important findings based on compelling evidence point to a potentially new role for dopamine signaling in the ventral striatum.

---

## [Decision Letter]

**Decision letter after peer review:**

Thank you for submitting your article "Ventral striatal dopamine encodes unique properties of visual stimuli in mice" for consideration by *eLife*. Your article has been reviewed by 3 peer reviewers, and the evaluation has been overseen by Geoffrey Schoenbaum as the Reviewing Editor and Kate Wassum as the Senior Editor. The reviewers have opted to remain anonymous.

Essential revisions:

In addition to the specific concerns of each reviewer, there were several things that were deemed essential and/or come out as themes across all reviewers, to which you should pay special attention. With your revision please include a point x point response to each point raise below and in the public review and recommendations for authors.

1) Some information and/or histology figure illustrating the placement of the fibers and also viral expression. Representative examples are needed as are schematics showing expression and placements for the group of subjects.

2) Additional discussion of the literature with regard to the specific choice of location within the ventral striatum.

3) Better integration or discussion of the current findings with the relevant literature that has come in recent years showing that dopamine can respond to events that are potentially independent of value. These are detailed in each reviewer's comments.

*Reviewer #1 (Recommendations for the authors):*

The scholarship of this study could be substantially improved, and some critical points need to be clarified or addressed:

– It is not immediately clear which mice were used for which experiments, what was the sensory history of each subject, or even what was the total number of mice used in the study. The authors mention that responses persisted after months of testing, so presumably, the same mice were tested and retested across different experiments. This information needs to be better detailed in the methods section, and it would be useful to have a visual timeline of each experiment.

– The authors do not provide any histological confirmation of fiber placement and viral expression. Not even a schematic of each fiber placement. This is a critical issue, especially as the lateral shell of the NAcc is a very small and narrow target in mice, and there are several discussions in the field about the potential computational specificity of dopaminergic signals in different NAcc compartments.

– The authors did not discuss or cite a number of relevant studies that have addressed similar issues, such as Kutlu et al. 2022 and Morrens et al. 2020, that have shown that dopamine transients in the VS evoked by visual, auditory, and odor cues are attenuated as a function of latent inhibition, similarly to some of results presented here. To what extent is the short-term sensory habituation found here different from this latent inhibition effect? Likewise, previous work has shown that dopamine neurons encode prediction errors generated by unexpected sensory (gustatory) transitions, including the specific sensory identity of the transitions (Takahashi et al., 2017; Stalnaker et al., 2019). Could the present results be a reflection of sensory prediction errors as proposed by these two previous papers?

– In the same vein as the previous point, one critical argument against the direct encoding of sensory stimuli properties by dopamine neurons or dopamine release is that these responses reflect a generalization of conditioning to previously rewarded events or contexts (Kobayashi and Schultz, 2014). According to this interpretation, dopamine is indeed only encoding reward-related information, but responses to neutral stimuli arise from a generalization of previous cue-reward pairings. If one were to apply this logic to the current study, it could be that the mice associate sharp transitions in their visual landscape with the renewal of their food supply (e.g., by experimenters or animal care staff opening the cage to give them more pellets), and therefore this is why there are sharp dopamine responses to these sensory events. Which components of the experiments in this study can rule out this interpretation, or at least mark it as unlikely?

References:

Kobayashi S, Schultz W (2014) Reward contexts extend dopamine signals to unrewarded stimuli. Curr Biol 24:56-62 Available at: http://dx.doi.org/10.1016/j.cub.2013.10.061.

Kutlu MG, Zachry JE, Melugin PR, Tat J, Cajigas S, Isiktas AU, Patel DD, Siciliano CA, Schoenbaum G, Sharpe MJ, Calipari ES (2022) Dopamine signaling in the nucleus accumbens core mediates latent inhibition. Nat Neurosci 25:1071-1081 Available at: https://pubmed.ncbi.nlm.nih.gov/35902648/ [Accessed September 29, 2022].

Morrens J, Aydin Ç, Janse van Rensburg A, Esquivelzeta Rabell J, Haesler S (2020) Cue-Evoked Dopamine Promotes Conditioned Responding during Learning. Neuron 106:142-153.e7.

Stalnaker TA, Howard JD, Takahashi YK, Gershman SJ, Kahnt T, Schoenbaum G (2019) Dopamine neuron ensembles signal the content of sensory prediction errors. *eLife* 8.

Takahashi YK, Batchelor HM, Liu B, Khanna A, Morales M, Schoenbaum G (2017) Dopamine Neurons Respond to Errors in the Prediction of Sensory Features of Expected Rewards. Neuron 95:1395-1405.e3 Available at: http://www.ncbi.nlm.nih.gov/pubmed/28910622 [Accessed October 24, 2017].

*Reviewer #2 (Recommendations for the authors):*

– One way to determine the role of stimulus prediction in canceling stimulus-evoked dopamine response would be to train mice to form tone → light associations and compare signaled (tone→light) vs unsignaled light-evoked dopamine responses. To be clear: I am not requesting the authors to perform this experiment. However, the potential role of stimulus predictability should be briefly discussed.

– Line 396: the authors mention that the inter-stimuli interval was 0.5ms. Judging by figure 1B it looks like the ISI might have to be 0.5s (5 stimuli of 1.1 s each, delivered over ~7.5 s, suggest an ISI of ~0.5s). This might have been a typo. Can the authors please check the ISI duration?

– The authors might want to include a video of the looming or receding discs (and their inverted version).

– One experiment measured tone-evoked dopamine responses. Was this done in the same mice previously used in visual experiments? If so, is there a correlation between light-evoked and tone-evoked dopamine responses?

*Reviewer #3 (Recommendations for the authors):*

Gonzalez and colleagues find that in the lateral ventral striatum, dopamine signals reliably report salient transitions in illuminance, scaling with light intensity and the speed of illuminance changes. They further find that the frequency of illuminance transitions, rather than the number per se, dictates the extent that dopamine signals habituate. In a number of studies, they characterize dopamine signals to light of different wavelengths, durations, and intensities. These results shed new light on the role of dopamine in signaling salience, independent of reward or threat learning. I thought the work was elegantly done and compellingly reported.

I have a few questions and comments.

A thought I had while reading this report is the question of why the lateral shell/ventral striatum was chosen as the focal point. There is, as the authors note, a reason to suggest that salience or value-free signals from dopamine occur in this portion of the striatum, but that is also the case in some studies for other regions, such as the accumbens core. I also appreciate that a previous paper from the senior author demonstrated a lateral shell visual cue-related effect. Nevertheless, I think the main limitation in the current data is a lack of comparison to another striatal region, either in the accumbens or elsewhere. Given the pretty extensive (and still growing) literature on dopamine heterogeneity in reward, valence, novelty, salience, and reinforcement signaling and function, I think it's reasonable to expect that some features of the demonstrated results here would differ, even in other ventral striatal regions. I would suggest including some discussion of this complexity, and perhaps changing the title to more specifically denote the findings here are in the lateral shell rather than "ventral striatum". Lateral, ventral, medial shell and core all have pretty well-documented variable dopamine dynamics to salient stimuli in other studies and that may play in here.

Conversely, a demonstration that at least some of the illuminance transition signals are similar across other striatal regions would be very interesting and informative in the context of the growing heterogeneity work. I'm not suggesting that the whole striatum needs to be probed, and definitely not to the level of detail as in the current report, but it would be really informative to know if some of the basics here hold/don't hold for another region.

In more specific questions, I was a little confused about the specific focus on dark-to-light transitions, given the framing of the evolved need to respond quickly to overhead threats, since that scenario presumably usually involves a light-to-dark transition (as a bird descends above the mouse). It wasn't clear if light-to-dark transitions were directly studied beyond the initial looming disc studies. Presumably, a rapid transition from bright lighting to darkness would also evoke strong dopamine.

I think it would be useful to emphasize in the descriptions of experiments for Figures 1 and 2 that long inter-stimulus intervals were used (I didn't learn this until the end of the methods). The authors later show that light transition signals don't really habituate if the ISIs are long – as I was reading I was confused about how rapidly repeated stimuli presentations were delivered, when intensity and duration conditions were within subject or between, etc. Perhaps a supplement figure showing more of the trial-by-trial data could be included to demonstrate a lack of habituation in the experiments in Figures 1 and 2 and clarify some of this.

The lack of clear habituation in the light transition signals when they occur separated by long intervals is super interesting. I do wonder if there are still possible long-term changes occurring, especially with mice that are getting 100s of light transition trials.

Related, do the authors know if auditory cue-evoked signals also don't habituate if spaced out? This isn't an experiment that would need to be done for this paper, but if that data exists it would be interesting to know if the habituation effect is similar across stimulus modalities or not.

---

## [Author Response]

Essential revisions:In addition to the specific concerns of each reviewer, there were several things that were deemed essential and/or come out as themes across all reviewers, to which you should pay special attention. With your revision please include a point x point response to each point raise below and in the public review and recommendations for authors.1) Some information and/or histology figure illustrating the placement of the fibers and also viral expression. Representative examples are needed as are schematics showing expression and placements for the group of subjects.

As mentioned in the response to Reviewer 1 below, we have added a new figure (Figure 1—figure supplement 1) that contains a representative confocal image showing the fiber location and virus expression, as well as the optical fiber locations. We apologize for this oversight in the original submission.

2) Additional discussion of the literature with regard to the specific choice of location within the ventral striatum.

We have added additional text stating our rationale for studying the lateral nucleus accumbens in the Introduction, as well as a more robust discussion of regional heterogeneity in dopamine responses to sensory stimuli throughout the striatum (Discussion, Paragraph 6). We believe that this is an important area of future study and have emphasized it accordingly.

3) Better integration or discussion of the current findings with the relevant literature that has come in recent years showing that dopamine can respond to events that are potentially independent of value. These are detailed in each reviewer's comments.

We would like to thank the reviewers for their helpful suggestions regarding previous literature examining environmental events independent of value. We have cited the suggested studies in the revised Discussion, as well as added new text directly addressing the reviewers’ concerns (Discussion, Paragraphs 2-3). This includes a new paragraph discussing our results in the context of sensory prediction error encoding (e.g. Takahashi et al., 2017; Stalnaker et al., 2019; Discussion, Paragraph 3). We hope that these actions do a better job acknowledging relevant foundational literature and providing context to the current findings. Details are provided in the responses to individual reviewers below.

Reviewer #1 (Recommendations for the authors):The scholarship of this study could be substantially improved, and some critical points need to be clarified or addressed:– It is not immediately clear which mice were used for which experiments, what was the sensory history of each subject, or even what was the total number of mice used in the study. The authors mention that responses persisted after months of testing, so presumably, the same mice were tested and retested across different experiments. This information needs to be better detailed in the methods section, and it would be useful to have a visual timeline of each experiment.

As mentioned above, we have better detailed the experimental workflow in the Materials and methods, as well as in a schematic presented in Figure 1-figure supplement 1. In order to alleviate concerns that a history of previous exposures could affect the response to repeated light stimuli, we have repeated critical experiments in a new cohort of stimulus-naïve mice, which is presented in the revised Figure 3.

– The authors do not provide any histological confirmation of fiber placement and viral expression. Not even a schematic of each fiber placement. This is a critical issue, especially as the lateral shell of the NAcc is a very small and narrow target in mice, and there are several discussions in the field about the potential computational specificity of dopaminergic signals in different NAcc compartments.

We apologize for this oversight. We have added both the individual fiber locations and a representative histological image to the manuscript (Figure 1—figure supplement 1). We have also included a discussion of regional heterogeneity of dopamine computations in ventral striatal subregions.

– The authors did not discuss or cite a number of relevant studies that have addressed similar issues, such as Kutlu et al. 2022 and Morrens et al. 2020, that have shown that dopamine transients in the VS evoked by visual, auditory, and odor cues are attenuated as a function of latent inhibition, similarly to some of results presented here. To what extent is the short-term sensory habituation found here different from this latent inhibition effect? Likewise, previous work has shown that dopamine neurons encode prediction errors generated by unexpected sensory (gustatory) transitions, including the specific sensory identity of the transitions (Takahashi et al., 2017; Stalnaker et al., 2019). Could the present results be a reflection of sensory prediction errors as proposed by these two previous papers?

We would like to thank the reviewer for their helpful comments regarding important foundational literature related to these studies. We have revised our Discussion section to include three new paragraphs of text to better place our findings in the context of previous studies. Topics include a discussion of dopaminergic encoding of sensory prediction errors (Takahashi et al., 2017; Stalnaker et al., 2019), the role of dopamine in latent inhibition (Kutlu et al., 2022), reward generalization, etc. If the reviewer has additional feedback on these changes, we would be happy to further revise the Discussion section.

– In the same vein as the previous point, one critical argument against the direct encoding of sensory stimuli properties by dopamine neurons or dopamine release is that these responses reflect a generalization of conditioning to previously rewarded events or contexts (Kobayashi and Schultz, 2014). According to this interpretation, dopamine is indeed only encoding reward-related information, but responses to neutral stimuli arise from a generalization of previous cue-reward pairings. If one were to apply this logic to the current study, it could be that the mice associate sharp transitions in their visual landscape with the renewal of their food supply (e.g., by experimenters or animal care staff opening the cage to give them more pellets), and therefore this is why there are sharp dopamine responses to these sensory events. Which components of the experiments in this study can rule out this interpretation, or at least mark it as unlikely?

This is an interesting point that we had not considered, which we now present in the Discussion section (Paragraph 2). While reward generalization could be a contributing factor to our findings, there are a few points that make it less likely. In our vivarium, the cages are changed during the light phase of the light-dark cycle, so a dark-to-light transition does not occur at cage opening. Additionally, food is always available to the mice, so they never undergo a period of fasting that would necessarily make opening of the cage a rewarding event. It is very possible, however, that there are rewarding stimuli associated with dark-to-light transitions that we have not accounted for, which could potentially affect the findings. Future studies will need to better parse this out.

Reviewer #2 (Recommendations for the authors):– One way to determine the role of stimulus prediction in canceling stimulus-evoked dopamine response would be to train mice to form tone → light associations and compare signaled (tone→light) vs unsignaled light-evoked dopamine responses. To be clear: I am not requesting the authors to perform this experiment. However, the potential role of stimulus predictability should be briefly discussed.

This is an excellent idea. Although investigating the role of dopaminergic responses to visual stimuli in behavioral conditioning is outside the scope of the current work, we are interested in pursuing this line of research in the future. To address the reviewer’s point, we have included new text in the Discussion related to stimulus predictability (Paragraph 3).

– Line 396: the authors mention that the inter-stimuli interval was 0.5ms. Judging by figure 1B it looks like the ISI might have to be 0.5s (5 stimuli of 1.1 s each, delivered over ~7.5 s, suggest an ISI of ~0.5s). This might have been a typo. Can the authors please check the ISI duration?

We apologize for the typo. The correct value is 0.5s. We have corrected the mistake. Thank you for pointing that out.

– The authors might want to include a video of the looming or receding discs (and their inverted version).

We have included a video showing the different stimulus types in the revised manuscript (Video 1).

– One experiment measured tone-evoked dopamine responses. Was this done in the same mice previously used in visual experiments? If so, is there a correlation between light-evoked and tone-evoked dopamine responses?

We have calculated the correlation between light and tone responses in mice exposed to both stimuli, and included the results in the source data file for Figure 2—figure supplement 1. There were no significant correlations between the light and tone responses for any frequency tested.

Reviewer #3 (Recommendations for the authors):Gonzalez and colleagues find that in the lateral ventral striatum, dopamine signals reliably report salient transitions in illuminance, scaling with light intensity and the speed of illuminance changes. They further find that the frequency of illuminance transitions, rather than the number per se, dictates the extent that dopamine signals habituate. In a number of studies, they characterize dopamine signals to light of different wavelengths, durations, and intensities. These results shed new light on the role of dopamine in signaling salience, independent of reward or threat learning. I thought the work was elegantly done and compellingly reported.I have a few questions and comments.A thought I had while reading this report is the question of why the lateral shell/ventral striatum was chosen as the focal point. There is, as the authors note, a reason to suggest that salience or value-free signals from dopamine occur in this portion of the striatum, but that is also the case in some studies for other regions, such as the accumbens core. I also appreciate that a previous paper from the senior author demonstrated a lateral shell visual cue-related effect. Nevertheless, I think the main limitation in the current data is a lack of comparison to another striatal region, either in the accumbens or elsewhere. Given the pretty extensive (and still growing) literature on dopamine heterogeneity in reward, valence, novelty, salience, and reinforcement signaling and function, I think it's reasonable to expect that some features of the demonstrated results here would differ, even in other ventral striatal regions. I would suggest including some discussion of this complexity, and perhaps changing the title to more specifically denote the findings here are in the lateral shell rather than "ventral striatum". Lateral, ventral, medial shell and core all have pretty well-documented variable dopamine dynamics to salient stimuli in other studies and that may play in here.Conversely, a demonstration that at least some of the illuminance transition signals are similar across other striatal regions would be very interesting and informative in the context of the growing heterogeneity work. I'm not suggesting that the whole striatum needs to be probed, and definitely not to the level of detail as in the current report, but it would be really informative to know if some of the basics here hold/don't hold for another region.

We agree with the reviewer that it is critical to know whether dopaminergic responses to visual stimuli generalize to other striatal sub-regions or show regional heterogeneity. This is a question we are actively exploring and hope to include it in a follow-up publication. As mentioned in the responses to other reviewers, we have greatly expanded our Discussion section to better highlight the issue of regional heterogeneity in the dopaminergic response to light, which I hope the reviewer finds addresses their concerns.

In more specific questions, I was a little confused about the specific focus on dark-to-light transitions, given the framing of the evolved need to respond quickly to overhead threats, since that scenario presumably usually involves a light-to-dark transition (as a bird descends above the mouse). It wasn't clear if light-to-dark transitions were directly studied beyond the initial looming disc studies. Presumably, a rapid transition from bright lighting to darkness would also evoke strong dopamine.

We focused on dark-to-light transitions because presentations of dark discs on a light background did not induce robust dopamine release (whereas light discs on dark backgrounds did). However, we believe that the reviewer’s point is important and would like to point to several places in the manuscript where dopamine responses to light-to-dark transitions were measured. For example, the dLight1 traces in Figure 2A, 3C, 3E, 4C, and 4E show the dopaminergic response to a rapid light-to-dark transition. In all cases, the dopaminergic ‘offset’ response is small (e.g. Figure 3C) and is dependent on the light stimulus duration (Figure 3E). This mimics light-adapted responses of OFF and ON-OFF retinal ganglion cells, although likely involves additional downstream neurocircuit mechanisms. While we did not test whether contrast inverted overhead fades (i.e. a fade out) could be encoded by lateral NAc dopamine release, the low amplitude dopamine response to black discs on a light background (e.g. the receding disc) suggests that it is not robust.

I think it would be useful to emphasize in the descriptions of experiments for Figures 1 and 2 that long inter-stimulus intervals were used (I didn't learn this until the end of the methods). The authors later show that light transition signals don't really habituate if the ISIs are long – as I was reading I was confused about how rapidly repeated stimuli presentations were delivered, when intensity and duration conditions were within subject or between, etc. Perhaps a supplement figure showing more of the trial-by-trial data could be included to demonstrate a lack of habituation in the experiments in Figures 1 and 2 and clarify some of this.

In order to address this concern, we have provided additional information about the stimulus parameters (stimulus duration, ISI, etc.) in the text related to Figure 2. We have also included seven trial-by-trial heatmaps (Figure 2—figure supplement 1) with associated peak data (Figure 2 —figure supplement 1 – source data 1) for white LED exposures across the entire irradiance range presented in Figure 2. Additionally, we have reworked our Methods section to clarify the stimulus parameters and trial structure for each experiment.

The lack of clear habituation in the light transition signals when they occur separated by long intervals is super interesting. I do wonder if there are still possible long-term changes occurring, especially with mice that are getting 100s of light transition trials.

As mentioned in the response to Reviewer 1, we have repeated the experiment in Figure 3D where mice were exposed to 100 s ISI light before and after three hundred 1 Hz light pulses in a new group of stimulus-naïve mice, which allowed us to be sure that a history of previous exposures did not affect our results. We then repeated the 100 s ISI stimulus train 48 hours later to determine if changes in the dopaminergic response to light after repeated exposure were durable. We found that the response white LED presentation was decreased by ~30% after 300 consecutive white LED exposures (1 second stimulus, 1 s ISI) within the same session; however, this reduction did not persist 48 hours later. These findings suggest that the dopaminergic response to light is relatively robust to previous exposure history between sessions.

Related, do the authors know if auditory cue-evoked signals also don't habituate if spaced out? This isn't an experiment that would need to be done for this paper, but if that data exists it would be interesting to know if the habituation effect is similar across stimulus modalities or not.

Given that responses to unconditioned tones were not robust in the LNAc in our hands, we did not ultimately pursue this line of research. However, it was recently shown by Dr. Erin Calipari’s group that dLight1 responses to neutral auditory stimuli in the nucleus accumbens core habituate with repeated exposures (see Kutlu et al., Nature Neuroscience, 2022). Examining the encoding of other sensory modalities by LNAc dopamine is an important future direction that we are considering going forward.